# Towards Effective 2-bit Quantization: Pareto-optimal Bit Allocation for Deep CNNs Compression

## Abstract

State-of-the-art quantization methods can compress deep neural networks down to 4 bits without losing accuracy. However, when it comes to 2 bits, the performance drop is still noticeable. One problem in these methods is that they assign equal bit rate to quantize weights and activations in all layers, which is not reasonable in the case of high rate compression (such as 2-bit quantization), as some of layers in deep neural networks are sensitive to quantization and performing coarse quantization on these layers can hurt the accuracy. In this paper, we address an important problem of how to optimize the bit allocation of weights and activations for deep CNNs compression. We first explore the additivity of output error caused by quantization and find that additivity property holds for deep neural networks which are continuously differentiable in the layers. Based on this observation, we formulate the optimal bit allocation problem of weights and activations in a joint framework and propose a very efficient method to solve the optimization problem via Lagrangian Formulation. Our method obtains excellent results on deep neural networks. It can compress deep CNN ResNet-50 down to 2 bits with only 0.7% accuracy loss. To the best of our knowledge, this is the first paper that reports 2-bit results on deep CNNs without hurting the accuracy.

## 1 Introduction

Deep Convolutional Neural Networks (CNNs) Krizhevsky et al. (2012); Simonyan & Zisserman (2015); Szegedy et al. (2015); He et al. (2016) have dominated the applications in computer vision and related fields. However, one challenging issue is that deep CNNs usually involve large number of parameters and are computationally demanding, making it very difficult to deploy deep CNNs on resource-limited devices. Quantization for weights and activations of deep CNNs is an effective way to mitigate this issue. Cooperated with advanced retraining (or fine-tuning) strategies, state-of-the-art methods Zhuang et al. (2018); Zhou et al. (2016); Zhang et al. (2018) can compress deep CNNs down to 4 bits without losing accuracy.

State-of-the-art quantization methods Zhuang et al. (2018); Zhou et al. (2016); Zhang et al. (2018) typically assign same bit rate to all layers for compression. However, when deep neural networks are compressed to very low precision, e.g., 2 bits, such equal bit allocation scheme is not reasonable. As some of layers in deep neural networks are sensitive to quantization, performing a coarse quantization on these layers may significantly hurt the accuracy. A better strategy is to adopt unequal bit allocation scheme to quantize weights and activations across layers.

In this paper, we address an important problem of how to optimize the bit allocation of weights and activations of deep CNNs. Finding the optimal bit allocation for deep CNNs is difficult, as traversing the whole searching space requires exponential computation complexity. Assume $M$ is the number of quantization levels and $L$ is the number of layers. A brute-force search has $O(M^L)$ computation complexity, which is unaffordable if $L$ is large in the case of deep neural networks. In our work, we first analyze how quantization affects the output of deep CNNs, and find that the mean squared output error caused by quantization has additivity property. Based on this observation, we formulate the optimal bit allocation problem of weights and activations in a joint framework

and adopt Lagrangian formulation Shoham & Gersho (1988) for optimization. We propose a very efficient method to find the solution with polynomial computation complexity.

Prior adaptive quantization schemes proposed by Khoram & Li (2018) and Zhou et al. (2018) only consider the compression of weights. Since the volume of activations is considerable, the size of activations should be also reduced as much as possible, which might otherwise slow down processing. Zhu et al. (2018) proposed layerwise adaptive quantization for weights and activations, but their method decides each layer's quantization level individually and does not optimize the bit allocation across layers when performing quantization on multiple layers. Different with prior literature, our work formulates the optimal bit allocation problem for both weights and activations and proposes an efficient method to solve the challenging problem. The new contributions of this paper are summarized as following:

- **Exploring Additivity of Mean Squared Output Error**: In this paper, we adopt mean squared output error to measure quantization impact. We observe that mean squared output error has additivity property, i.e., the mean squared output error caused by quantizing multiple layers equals to the sum of mean squared output error due to the quantization of each individual layer. We show empirical results on two deep CNNs AlexNet and VGG-16 and also provide a mathematical derivation for the additivity property.

- **Pareto Condition for Optimization**: Based on the additivity of mean squared output error, we propose an optimal bit allocation framework for both weights and activations. Moreover, we propose an efficient method to solve the optimization problem by using Lagrangian Formulation. We show that the optimization problem can be solved in polynomial computational complexity under Pareto condition, i.e., the slopes of mean squared output error vs bit rate functions of each layer have to be equal.

- **Impact on Improving Inference Rate**: The pattern of Pareto-optimal bit allocation of weights on deep CNNs has very positive impacts on inference rate. It tends to allocate fewer bits per weight for layers that have a lot of weights, which helps to reduce the corresponding memory-access time which in turn reduces compute idle time and improves the overall inference rate. Simulation experiments on Google TPU hardware accelerator shows that Pareto-optimal bit allocation can further improve the inference rate on deep CNNs by 1.5X compared to its equal bit allocation counterpart.

Results on ImageNet Deng et al. (2009) show that our method can compress deep CNNs (e.g., ResNet-50 He et al. (2016)) down to 2 bits and the accuracy loss is less than $0.7\%$. To the best of our knowledge, this is the first paper that reports 2-bit results on deep CNNs without hurting the accuracy. The remainder of this paper is structured as follows. Section 2 discusses related works. Section 3 analyzes the impact of quantization. Section 4 develops our bit allocation framework. Section 5 reports experimental results. Section 6 discusses the impact of inference rate.

## 2 RELATED WORKS

Some latest works Zhuang et al. (2018); Park et al. (2018); Choi et al. (2018b;a) already reported 2-bit results on deep CNNs. Among them, the highest result is reported by Park et al. (2018), which loses 2.45% accuracy at 2 bits on ResNet-50 He et al. (2016). Although promising results are obtained, there is still considerable accuracy drop at 2 bits. Note that all these works assign same bit rate to quantize weights and activations in different layers. Different with them, our method adopts a better strategy for bit allocation. The optimal bit allocation strategy is critical for maintaining high accuracy when neural networks are compressed to very low precision like 2 bits.

The most relevant works related to our unequal bit allocation scheme are proposed by Khoram & Li (2018) and Zhou et al. (2018). Khoram & Li (2018) suggest monitoring the loss function during training to allocate the bit rates of weights in different layers. Zhou et al. (2018) suggest applying adversarial noise Fawzi et al. (2016) as an indicator to measure the quantization impact for bit allocation. Both of these two works only study the quantization problem of weights.

Zhou et al. (2018) empirically finds that the adversarial noise Fawzi et al. (2016) has additivity property. While, adversarial noise is a measurement of top-1 accuracy in object classification task, which is not applicable to other tasks like object detection. Our work studies the additivity of mean

Figure 1: An example of uniform quantizer with dead zone.

squared output error. We do not make assumptions about the application scenarios of our method We provide both empirical results and a mathematical derivation for the additivity property.

Banner et al. (2019) proposed per-channel bit allocation for weights and activations. They formulate the bit allocation problem as a minimization problem of Mean Squared Error (MSE) of quantized values in intermedia layers, and provide a solution with analytic expression in the case of Laplacian distribution. Finally, the solution of bit allocation with analytic expression is rounded to integer values. Mean Squared Error (MSE) in intermedia layers can not reflect network networks' final output accuracy. Our method adopts mean squared output error to measure the impact on last layer's output accuracy. It finds the optimal bit allocation directly from the discrete search space and does not make any assumptions for the distributions.

Recently, Zhe et al. (2019) adopted the rate-distortion Lagrangian formulation to allocate bitrates for weights and activations across layers. Our method builds on top of the Lagrangian formulation approach. Beyond that, we provide the analysis of the mathematical derivation of additivity property, which is the precondition of Lagrangian formulation. The proposed quantization framework differs from Zhe et al. (2019) in two-fold. First, we adopt a dead zone to the quantization function of weights. Second, we apply the straight-through estimator (STE) Bengio et al. (2013) to perform back-propagation on the retraining stage for both quantized weights and activations. As illustrated in the experiment section, dead zone and STE retraining are critical for improving the accuracy. In addition, we show that the bit allocation solved by our approach has positive impacts at inference stage. It can largely speed up the inference rate compared to its equal bit allocation counterpart. We verify this point by performing hardware simulation experiments on Google TPU v1 platform.

## 3 ANALYSIS OF QUANTIZATION IMPACT

We adopt uniform scalar quantization in our compression framework. Each layer's weights and activations are quantized separately, followed by entropy coding of the quantization index. The quantizer stepsize is individually chosen for each layer's weights and activations.

We incorporate a dead zone into the quantizers for weights. Let $\Delta$ denote the stepsize of a quantizer. The width of the dead zone is $\Delta + \beta$. Let $t = 2k + 1$ denote the total number of representative levels of the quantizer. Given scalar input $x$, if $x$ is in dead zone ($|w| < \frac{\Delta + \beta}{2}$), it is quantized to zero. Otherwise, $x$ is quantized to

$$q(x) = sign(x) \cdot \left( \frac{\beta}{2} + |i| \cdot \Delta \right), \tag{1}$$

where integer $i$ is the quantization index, and $1 \leq |i| \leq k$. Both $\beta$ and $t$ are hyper parameters. We note that adding a dead zone can improve compression. This will be discussed in the following sections.

We do not apply a dead zone to the quantizers of activations. We assume activation function after Convolution layer is ReLU, which is common practice in modern deep CNNs Krizhevsky et al. (2012); Simonyan & Zisserman (2015); He et al. (2016). The quantizer for activations is define as

$$q(x) = i \cdot \Delta, \tag{2}$$

where quantization index $0 \leq i \leq t - 1$.

### 3.1 MEAN SQUARED OUTPUT ERROR CAUSED BY QUANTIZATION

Let $W_i$ denote weights in layer $i$ and $A_i$ denote activations in layer $i$ for all $1 \leq i \leq L$ where $L$ is total number of layers. Given a neural network $\mathcal{F}$ and input image $I$, we have output vector $V = \mathcal{F}(I)$. If we quantize the weights $W_i$, we obtain a modified output vector $\widehat{V}$. The mean sqaured

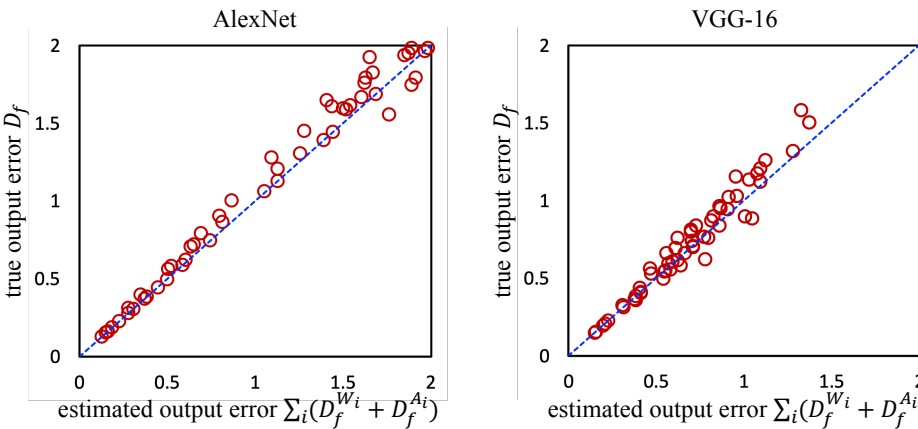

Figure 2: Demonstration of the additivity of mean squared output error on AlexNet and VGG-16. For each time, we randomly select some of layers and perform quantization on the selected layers, which corresponds a point in the figures.

output error caused by quantizing $W_i$ is defined as the expectation of the squared Euclidean distance between original output $V$ and modified output $\widehat{V}$ divided by the dimensionality of $V$

$$D_{\mathcal{F}}^{W_i} = \frac{E(d(V, \widehat{V}))}{dim(V)}, \tag{3}$$

where $E(.)$ denotes the expectation operator and $d(X, Y)$ is the squared Euclidean distance between vectors $X$ and $Y$. The expectation is over the distribution of all random input images $I$. Similarly, $D_{\mathcal{F}}^{A_i}$ denotes mean squared output error caused by quantization of activations $A_i$ for all $1 \leq i \leq L$. In the following, "mean-squared output error" and "output error" will both be used interchangeably.

## 3.2 ADDITIVITY OF MEAN SQUARED OUTPUT ERROR

Consider mean squared output error when simultaneously quantizing all layers. Let $D_{\mathcal{F}}$ denote mean squared output error caused by quantizing all layer's weights and activations, the following additivity property holds.

**Proposition 1** *The output error $D_{\mathcal{F}}$, caused by quantizing all layers' weights and activations, equals the sum of all output error due to the quantization of individual layer's weights and activations*

$$D_{\mathcal{F}} = D_{\mathcal{F}}^{W_1} + ... + D_{\mathcal{F}}^{W_L} + D_{\mathcal{F}}^{A_1} + ... + D_{\mathcal{F}}^{A_L}, \tag{4}$$

*if the neural network is continuously differentiable in every layer and quantization errors in different layers are independently distributed with zero mean.*

In practice, additivity holds approximately in deep neural networks. We evaluate additivity on AlexNet and VGG-16. Figure 2 shows the results. Vertical axis in Figure 2 represents the output error when quantizing all layers simultaneously. Horizontal axis represents the sum of output error caused by quantizing each layer individually. We can see that data points in Figure 2 are all very close to the diagonal, meaning that additivity property holds. This property can be shown mathematically by linearizing the quantization error of weights and activations using Taylor series expansion as the neural networks is continuously differentiable and the quantization error can be considered as small deviations. The supplementary material provides a mathematical derivation.

## 4 UNEQUAL BIT ALLOCATION FRAMEWORK

Let $R^{W_i}$ and $R^{A_i}$ denote the total bitrates of weights and activations in i-th layer, respectively. $R^{W_i}$ (or $R^{A_i}$) is determined by the number of representative levels $t$, the dead zone parameter $\beta$, and

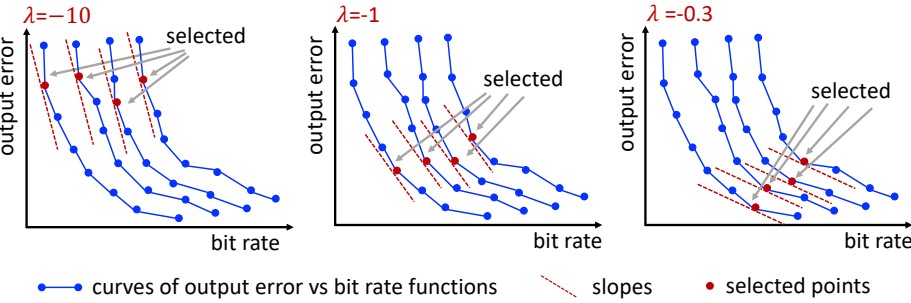

Figure 3: Example of finding optimal bit allocation under Pareto condition (best viewed in color).

the number of weights (or activations) in layer $i$. According to 4, the output error of quantizing all layers' weights and activations is the sum of the output error caused by quantizing each layer's weights and activations individually. The bit allocation framework is thus formulated as

$$\arg\min_{R^{W_i}, R^{A_i}} \quad D_{\mathcal{F}}^{W_1} + D_{\mathcal{F}}^{A_1} + ... + D_{\mathcal{F}}^{W_L} + D_{\mathcal{F}}^{A_L}$$

$$s.t. \quad R^{W_1} + R^{A_1} + ... + R^{W_L} + R^{A_L} = R^{Total}, \tag{5}$$

where $R^{Total}$ is the total number of bits needed to represent the weights and activations of all layers. Our goal is to find the optimum bit allocation $\{R^{W_i}\}_{i=1}^{L}$ and $\{R^{A_i}\}_{i=1}^{L}$ to minimize network output error within a constrained total bit budget for weights and activations. More precisely, we adjust representative level $t$ and dead zone parameter $\beta$ of the quantizer in each layer to minimize the overall output error of the quantized network given a total bit budget constraint.

### 4.1 Pareto-optimal Bit Allocation

We use a classical Lagrangian rate-distortion formulation to find the optimum bit allocation Shoham & Gersho (1988). Our Lagrangian cost function is

$$\mathcal{J} = \sum_i (D_{\mathcal{F}}^{W_i} + D_{\mathcal{F}}^{A_i}) - \lambda \cdot \sum_i \left( R^{W_i} + R^{A_i} \right). \tag{6}$$

By setting the partial derivatives of $\mathcal{J}$ with respect to $R^{W_i}$ and $R^{A_j}$ to zero, we obtain

$$\frac{\partial D_{\mathcal{F}}^{W_i}}{\partial R^{W_i}} = \frac{\partial D_{\mathcal{F}}^{A_j}}{\partial R^{A_j}} = \lambda, \tag{7}$$

for all $1 \leq i, j \leq L$. The above equation expresses the intuitively pleasing insight that the slopes of output error versus rate functions for weights and activations of each layer have to be equal. This is exactly the well-known Pareto condition for optimal resource allocation in economics. 7 holds, as long as the output error contributions $D_{\mathcal{F}}^{W_i}$ and $D_{\mathcal{F}}^{A_i}$ are additive, rates $R^{W_i}$ and $R^{A_i}$ are additive, the functions $D_{\mathcal{F}}^{W_i}(R^{W_i})$ and $D_{\mathcal{F}}^{A_i}(R^{A_i})$ are convex and the resulting $R^{W_i}, R^{A_i} > 0$. It does not make assumptions about the specific quantization or coding scheme used.

### 4.2 Optimization under Pareto Condition

We find optimum bit allocation under Pareto condition according to 7. We first generate rate-distortion curves of $(R^{W_i}, D_{\mathcal{F}}^{W_i})$ and $(R^{A_i}, D_{\mathcal{F}}^{A_i})$ by choosing different values for representative level $t$ and dead zone parameter $\beta$. Then we enumerate $\lambda$ and select the point in each curve with slope equal to $\lambda$. The set of selected points is one solution of Pareto-optimal bit allocation. We may enumerate many values of $\lambda$ to find the best solution with minimal output error under the bit budget constraint. Assume that we have $N$ curves and $M$ points in each curve. The whole search space of the solution is $O(M^N)$. By utilizing the Pareto condition, the time complexity to find optimum bit allocation is reduced to $O(K \cdot M \cdot N)$, where $K$ is the total number of $\lambda$ evaluated. In practice, we can find the optimum bit allocation for ResNet-50 He et al. (2016) and MobileNet-v1 Howard et al.

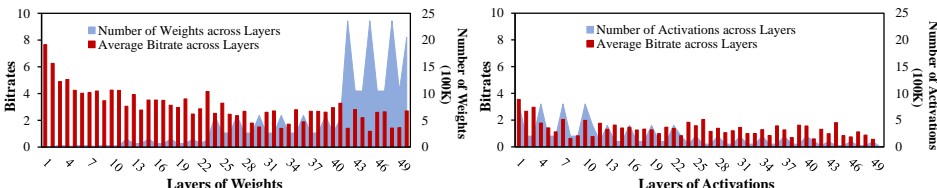

Figure 4: Pareto-optimal bit allocation of weights and activations across layers on ResNet-50 when compressed into 2 bits on average.

(2017) in several seconds on a standard CPU (AMD Ryzen threadripper 1950x) using the algorithm described above.

Figure 3 illustrates the optimization steps under Pareto condition. Each blue curve in Figure 3 denotes a rate-distortion curve in one layer. We totally show 4 rate-distortion curves in Figure 3. The optimization starts from enumerating the value of slope $\lambda$. Given $\lambda$ (dotted red lines in Figure 3), we then find the point with slope equal to $\lambda$ for each curve. Specifically, we compute the intercepts on Y-axis of the lines with slope equal to $\lambda$, which pass the points on the curve. The point with minimal intercept on Y-axis will be selected as solution. Note that we only select one point with minimal Y-axis intercept for each curve. The selected point corresponds to the bitrate in the related layer. We start $\lambda$ from a very small value and increase $\lambda$ gradually until the total rate reaches the target. The example in Figure $\lambda$ totally evaluates three different values of $\lambda$.

### 4.3 Observations on Optimum Bit Allocation

Figure 4 shows the optimum bit allocation of weights and activations according to 7 on ResNet-50 He et al. (2016), when weights and activations are compressed to 2 bits on average. Two observations stand out. First, weights receive much larger bitrates (average bits per layer) than activations in general. Second, the layers with larger number of weights receive relatively lower bitrates; conversely, the layers with small number of weights receive relatively high bitrates. Similar trends are observed on MobileNet-v1 Howard et al. (2017), which are illustrated in the supplementary material. We noticed that such bit allocation pattern of weights can have a positive impact on inference rate. As we allocate relatively small bitrates to large-weight layers, the corresponding memory-access time will be reduced which in turn reduces compute idle time and improves the overall inference rate (see Section 6).

## 5 Experiments

We evaluate our method on ResNet-50 He et al. (2016) and MobileNet-v1 Howard et al. (2017) over the ImageNet Deng et al. (2009) validation dataset with 224x224 input image resolution. The total number of weights and activations of ResNet-50 is 32.3M with 22.7M weights and 9.6M activations, and the total number of weights and activations of MobileNet-v1 is 9.3M with 4.2M weights and 5.0M activations. We adopt the Tensorflow implementation of ResNet-50 and MobileNet-v1 provided by the Tensorpack and SLIM library, respectively. The original ResNet-50 and MobileNet-v1 have 76.4% and 70.9% top-1 image classification accuracy on the ImageNet validation dataset, respectively.

We compare our method with 15 state-of-the-art methods, including Binarized Weight Network (BWN) Hubara et al. (2016), Ternary Weight Network (TWN) Li et al. (2016), Incremental Network Quantization (INQ) Zhou et al. (2017), Fine-grained Quantization (FGQ) Mellempudi et al. (2017), Integer Arithmetic-only Inference (IAOI) Jacob et al. (2018), Adaptive Quantization (AQ) Zhou et al. (2018), Compression Learning by In-parallel Quantization (CLIP-Q) Tung & Mori (2018), Symmetric Quantization (SYQ) Faraone et al. (2018), Low-bitwidth CNN (LBCNN) Zhuang et al. (2018), Extremely Low Bit Neural Network (ELBNN) Leng et al. (2017), Learned Quantization (LQ-Nets) Zhang et al. (2018), Deep Compression (DC) Han et al. (2016), Hardware-Aware Automated Quantization (HAQ) Wang et al. (2019), Parameterized Clipping Activation (PACT) Choi et al. (2018b), DoReFa-Net Zhou et al. (2016), and Joint Bit Allocation (JBA) Zhe et al. (2019). It is

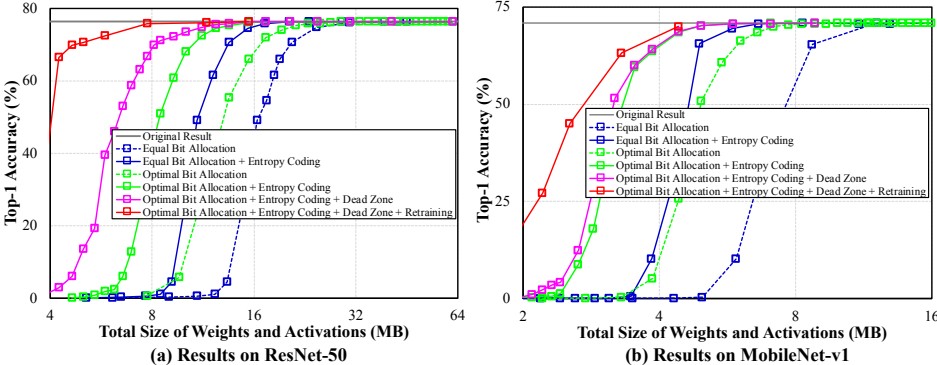

Figure 5: Effectiveness of each component in the proposed compression framework. Results are shown over the ImageNet validation dataset when weights and activations are compressed to different sizes. The size of original ResNet-50 and MobileNet-v1 are 116 MB and 35 MB, respectively.

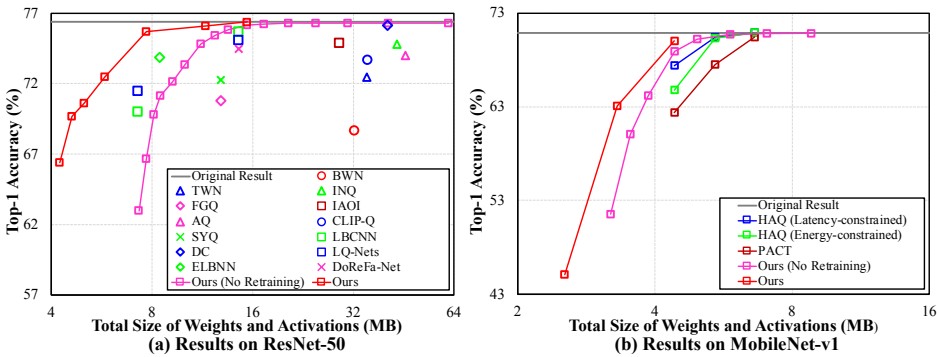

Figure 6: Tradeoff between size and accuracy on ResNet-50 and MobileNet-v1 over ImageNet validation dataset comparing with state-of-thhe-art methods from the literature.

worth noting that the top performing comparison methods all include retraining to recover accuracy loss due to quantization error introduced.

## 5.1 EFFECTIVENESS OF THE PROPOSED COMPRESSION FRAMEWORK

In this section, we evaluate the effectiveness of each component in our compression framework.

**Optimal Bit Allocation.** As illustrated in Figure 5, we show the performance of equal bit allocation as a baseline to evaluate the effectiveness of the proposed optimal bit allocation scheme. Equal bit allocation assigns same quantization level for all layers' weights and activations. In Figure 5, there is a large gap between the results of optimal bit allocation (green curves) and the results of equal bit allocation (blue curves). Optimal bit allocation is much more effective.

**Entropy Coding and Dead Zone.** Applying entropy coding to quantization index can further reduce the size by 20% to 30%. We use the entropy of the quantization index to estimate the code size after entropy coding. In Figure 5, the dashed green and blue curves show the results of quantized networks with equal and optimal bit allocation before entropy coding, respectively. After entropy coding, the size of quantized networks can be further reduced (see the solid green and blue curves in Figure 5). Besides, with a dead zone, we also observe large performance improvement on ResNet-50 and noticeable improvement on MobileNet-v1. Adding a dead zone to the quantizers forces more weights to zero. This benefits entropy coding, as it makes the distribution of quantized weights more peaky. A dead zone is similar to pruning Han et al. (2015; 2016); both set small weights to zero. The key difference is pruning completely removes the small weights from networks while dead zone still keeps them.

Table 1: Comparing our method with state-of-the-art methods on ResNet-50 and MobileNet-v1 over ImageNet validation dataset when weights and activations are compressed to low bitrates.

| Models | Methods | Original | | 2 Bits | | 4 Bits | |
|---|---|---|---|---|---|---|---|
| | | Top 1 | Top 5 | Top 1 | Top 5 | Top 1 | Top 5 |
| ResNet-50 | DoReFa-Net Zhou et al. (2016) | 75.6 | 92.2 | 67.3 | 84.3 | 74.5 | 91.5 |
| | LBCNN Zhuang et al. (2018) | 75.6 | 92.2 | 70.0 | 87.5 | 75.7 | 92.0 |
| | LQ-Nets Zhang et al. (2018) | 76.4 | 93.2 | 71.5 | 90.3 | 75.1 | 92.4 |
| | JBA Zhe et al. (2019) | 75.2 | 92.2 | - | - | 74.2 | 91.5 |
| | Ours (No Retraining) | 76.4 | 93.2 | 66.7 | 87.5 | **76.2** | **93.1** |
| | Ours | 76.4 | 93.2 | **75.7** | **92.8** | **76.4** | **93.3** |
| MobileNet-v1 | PACT Choi et al. (2018b) | 70.8 | 89.9 | - | - | 62.4 | 84.2 |
| | HAQ (energy) Wang et al. (2019) | 70.8 | 89.9 | - | - | 64.8 | 85.9 |
| | HAQ (latency) Wang et al. (2019) | 70.8 | 89.9 | - | - | 67.5 | 87.9 |
| | Ours (no retraining) | 70.8 | 89.9 | 2.1 | 6.0 | **68.9** | **88.6** |
| | Ours | 70.9 | 89.9 | 27.3 | 50.7 | **70.1** | **89.2** |

**Retraining.** Like other schemes, our compression framework can be enhanced by retraining, especially at very low bitrates. We adopt the straight-through estimator (ETS) Bengio et al. (2013) to perform back-propagation through non-differentiable quantization functions. We retrain 10 epochs for both ResNet-50 He et al. (2016) and MobileNet-v1 Howard et al. (2017). For ResNet-50, we set the learning rate as 0.1 if bitrate is more than 3 bits, otherwise, learning rate is 0.001. For MobileNet-v1 Howard et al. (2017), learning rate is 0.45. Without retraining, our method obtains 66.7% on ResNet-50 at 2 bits on average. The simple retraining stage improves accuracy to 75.7%—close to that of the original pre-trained model (76.4%).

## 5.2 COMPARISON WITH STATE-OF-THE-ART METHODS

The proposed compression framework achieves excellent performance on ResNet-50 and MobileNet-v1, compared to recent state-of-the-art methods. Figure 6 shows the tradeoff between size of weights/activations and classification accuracy over the ImageNet validation dataset. On MobileNet-v1, our method without retraining outperforms state-of-the-art approaches equipped with retraining. Similar observation can be made for ResNet-50.

Table 1 further compares our method with state-of-the-art that report results at very low bitrates (average 2 and 4 bits). At 4-bit on average, without retraining, our method only loses 0.2% Top-1 accuracy on ResNet-50. This is strong evidence of the effectiveness of the proposed compression framework. With simple retraining, our method shows additional accuracy gains. On ResNet-50, it outperforms the best performing baseline by 4.2% Top-1 accuracy at 2 bits. On MobileNet-v1, our method outperform the best performing baseline by 2.6% Top-1 accuracy at 4 bits. Further improvements are possible if combining our unequal bit allocation scheme with advanced quantized model retraining approaches like progressive training and knowledge distillation training Zhuang et al. (2018).

We compare our method with the very recent work JBA Zhe et al. (2019). At 4 bits on ResNet-50, our method without retraining outperforms JBA by 2.0% (76.2% vs 74.2). As observed in Figure 3 from the paper Zhe et al. (2019), the results of JBA Zhe et al. (2019) drops more than 15% at 2 bits. While, our method without retraining drops 9.7% accuracy at 2 bits, which significantly outperforms JBA, which is attributed to the dead zone quantization. With retraining, our method only loses 0.7% accuracy at 2 bits, outperforming JBA by more than 10.0%.

We also report our results at 1.5 bits. We noticed that the quantized networks with equal bit allocation at 2 bits without entropy coding can be further compressed to around 1.5 bits after entropy coding. Our method obtains 72.5 Top-1 accuracy at 1.5 bits on ResNet-50, which still outperforms other state-of-the-art methods at 2 bits by 1.0%.

Table 2: Architecture parameters of considered deep learning hardware platforms.

| Paramter | MIT Eyeriss | Google TPU |
|---|---|---|
| On-chip memory | 181.5 KBytes | 28 MBytes |
| Off-chip memory-access bandwidth (BW) | 1 GByte/sec | 13 GBytes/sec |
| Computing performance (Perf) | 34 GOPs | 96 TOPs |

Table 3: Inference rates of equal and unequal bit allocation schemes, as well as full precision, on ResNet-50 and MobileNet-v1. TPU uses 8-bit precision while MIT Eyeriss uses 16-bit precision.

| Platform | Model | 2 Bits | | | 4 Bits | | | Speedup vs. full precision | |
|---|---|---|---|---|---|---|---|---|---|
| | | equal | ours | speedup | equal | ours | speedup | 2 bits | 4 bits |
| TPU | ResNet | 833 | 1250 | 1.5x | 625 | 666 | 1.07x | 4x | 2.1x |
| | Mobile | 892 | 1175 | 1.3x | 979 | 979 | 1x | 1.8x | 1.5x |
| Eyeriss | ResNet | 8 | 8 | 1x | 8 | 8 | 1x | 1.05x | 1.05x |
| | Mobile | 12 | 12 | 1x | 12 | 12 | 1x | 1.05x | 1.05x |

## 6 DISCUSSIONS

### 6.1 ANALYSIS OF COMPUTATIONAL COMPLEXITY

We compare the computational complexity of our method and the equal bit allocation method, by counting the number of arithmetic operations required for neural network inference with batch size 1. Following the protocol defined by MicroNet challenge [1], we consider a 32-bit multiplication/addition as one operation. For mixed-precision computations (e.g., 3 bit weight times 5 bit input), we count the number of of operations to be the maximum bit-width of the 2 operands of this operation, divided by 32. For example, a multiplication operation between a 3-bit and a 5-bit operand will count as 5/32 of an operation.

On ResNet-50, our method has less amount of operations than the equal bit allocation method when weights and activations are compressed to 4 bits ($7.0 \times 10^8$ vs $7.2 \times 10^8$) and 6 bits ($9.4 \times 10^8$ vs $10.8 \times 10^8$). At 2 bits, our method has 1.4x more operations than the equal bit allocation method ($5.1 \times 10^8$ vs $3.6 \times 10^8$).

In practice, the inference time on hardware platforms is constrained by both compute and memory access. The Pareto-optimal bit allocation tends to allocate fewer bits per weight for layers that have a lot of weights, which helps to reduce the corresponding memory-access time which in turn reduces compute idle time and improves the overall inference speed. We show the simulation results on hardware platforms in the following section.

### 6.2 IMPACT ON INFERENCE RATE

We define inference rate as the maximum number of images that a particular implementation of a neural network can process per unit time. Inference rate is mainly driven by memory-access bandwidth and compute throughput (number of compute operations per unit time). In our work, entropy-encoded weights and activations are fetched from off-chip to on-chip memory, where they are decoded and fed into the processing units.

We have explored the impact our approach has on the inference rate for different hardware architectures inspired by current embedded and high-performance DNN-targeted hardware accelerators (i.e., MIT Eyeriss Chen et al. (2016) and Google TPU Jouppi et al. (2017), summarized in Table 2)—such platforms have shown superior inference rate compared to GPUs Jouppi et al. (2017). We assume that accessing off-chip memory can be done in parallel while computing is performed (i.e., weights can be prefetched from off-chip memory).

Using SCALE-sim Samajdar et al. (2018) software, we simulate the inference rate of ResNet-50 and MobileNet-v1, when mapped into the two considered hardware platforms in Table 2. We report the

---

[1]`https://micronet-challenge.github.io`

inference speed-up resulting from our unequal bit allocation scheme versus the equal bit allocation scheme (all weights and activations are represented with a fixed number of bits), when running an inference on a single $224 \times 224$ input image. We also report the speedup compared to a full precision allowed on each platform (i.e., 8-bit on TPU, 16-bit on Eyeriss).

Our unequal bit allocation achieves $1.5\times$ and $1.07\times$ higher inference rate for ResNet-50 ($1.3\times$ and $1\times$ for MobileNet-v1) at 2 bits and 4 bits, as summarized in Table 3, when compared with their equal bit allocation counterparts, respectively. Higher speed-ups, up to $4\times$, are achieved when compared with 8-bit full precision. One may note that the speedup is much larger if we consider inference time contributed by memory access only.

These improvements are mainly attributed to the unequal bit allocation pattern—it minimizes the layer-wise bitrates for weights/activations and also balances the size of data moving between off-chip and on-chip memory, which leads to reduced access times to off-chip memory. It is worth mentioning that with higher memory-access bandwidth, lower speed-ups are observed. For instance, at 128 GBytes/s, no speedup is observed. However, such hardware modules consume high energy and are rather costly, where are our approach provides an attractive alternative to increased memory bandwidth. Furthermore, platforms with low computation rate (i.e., few GOPs, as in the case of Eyeriss in Table 2), are constrained by the computational throughput, hence may not benefit from any speed-up.

## 7 CONCLUSIONS

In this paper, we have proposed a novel unequal bit allocation framework for compression of both weights and activations of deep CNNs. The output error due to the quantization of individual layer's weights and activations is additive, as are the bits used to represent each layer. Building on this observation, we formulate a Lagrangian optimization framework to find the optimum bit allocation. The optimal bit allocation problem can be efficiently solved by utilizing the Pareto condition, reducing the search space from exponential complexity to linear complexity. Our method achieves excellent results on ResNet-50 and MobileNet-v1. In particular, our method can compress ResNet-50 into 4 bits on average without retraining, which is computationally very expensive and might not even be possible, if the original training data are not available. The proposed unequal bit allocation scheme has a very positive impact on inference rate. It is able to improve inference rate by up to $4\times$ compared with full precision, as well as $1.5\times$ compared with its equal bit allocation counterpart, on Google TPU hardware accelerator.

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

# SUPPLEMENTARY MATERIAL
# TOWARDS EFFECTIVE 2-BIT QUANTIZATION: PARETO-OPTIMAL BIT ALLOCATION FOR DEEP CNNS COMPRESSION

**Anonymous authors**

## ABSTRACT

In this supplementary material, we provide the mathematical analysis for the additivity of output error, and the additional results of Pareto-optimal bit allocation. Section 1 provides the proof of Proposition 1. Section 2 provides the results of Pareto-optimal bit allocation on ResNet-50 He et al. (2016) and MobileNet-v1 Howard et al. (2017) at different bitrates.

## 1 ADDITIVITY OF OUTPUT ERROR

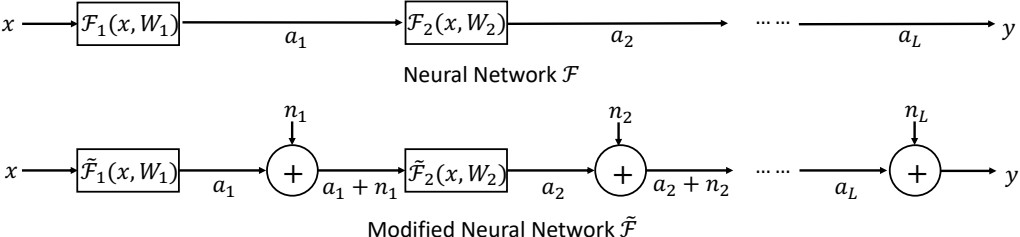

Figure 1: Examples of a neural network $\mathcal{F}$ and a modified neural network $\widetilde{\mathcal{F}}$.

**Proposition 1** *The output error $D_{\mathcal{F}}$, caused by quantizing all layers' weights and activations, equals the sum of all output error due to the quantization of individual layer's weights and activations*

$$D_{\mathcal{F}} = D_{\mathcal{F}}^{W_1} + ... + D_{\mathcal{F}}^{W_L} + D_{\mathcal{F}}^{A_1} + ... + D_{\mathcal{F}}^{A_L} \tag{1}$$

*if the neural network is continuously differentiable in every layer, the quantization errors in different layers are independently distributed with zero mean.*

**Proof 1** *Let $\mathcal{F}(W_1, ..., W_L)$ denote a neural networks and $\widetilde{\mathcal{F}}(W_1, ..., W_L, n_1, ..., n_L)$ denote a modified neural networks of $\mathcal{F}$ where an element-wise add layer with parameter $n_i$ is followed for each activation $a_i$. Based on this definition, we have*

$$\mathcal{F}(W_1, ..., W_L) = \widetilde{\mathcal{F}}(W_1, ..., W_L, 0, ..., 0). \tag{2}$$

*Let $X_0 = (W_1, ..., W_L, 0, ..., 0)$ and $\Delta X = (\Delta W_1, ..., \Delta W_L, \Delta n_1, ..., \Delta n_1)$. Assume that the quantization error can be considered as small deviation. We apply the Taylor series expansion up to first order term on $\widetilde{\mathcal{F}}$ at $X_0$,*

$$\widetilde{\mathcal{F}}(X_0 + \Delta X) - \widetilde{\mathcal{F}}(X_0) = \sum_{i=1}^{L} \frac{\partial \widetilde{\mathcal{F}}}{\partial W_i} \cdot \Delta W_i + \sum_{i=1}^{L} \frac{\partial \widetilde{\mathcal{F}}}{\partial n_i} \cdot \Delta n_i. \tag{3}$$

*Then, $\|\widetilde{\mathcal{F}}(X_0 + \Delta X) - \widetilde{\mathcal{F}}(X_0)\|^2$ can be written as*

$$\Big( \sum_{i=1}^{L} \Delta W_i{}^\top \cdot \frac{\partial \widetilde{\mathcal{F}}}{\partial W_i}{}^\top + \sum_{i=1}^{L} \Delta n_i{}^\top \cdot \frac{\partial \widetilde{\mathcal{F}}}{\partial n_i}{}^\top \Big) \cdot \Big( \sum_{i=1}^{L} \frac{\partial \widetilde{\mathcal{F}}}{\partial W_i} \cdot \Delta W_i + \sum_{i=1}^{L} \frac{\partial \widetilde{\mathcal{F}}}{\partial n_i} \cdot \Delta n_i \Big). \quad (4)$$

*Because we assume that quantization errors in different layers are independently distributed with zero mean, the cross terms of (4) disappear when taking the expectation. That is:*

$$E(\Delta W_j{}^\top \cdot \frac{\partial \widetilde{\mathcal{F}}}{\partial W_j}{}^\top \cdot \frac{\partial \widetilde{\mathcal{F}}}{\partial W_i} \cdot \Delta W_i) = E(\Delta W_j{}^\top) \cdot \frac{\partial \widetilde{\mathcal{F}}}{\partial W_j}{}^\top \cdot \frac{\partial \widetilde{\mathcal{F}}}{\partial W_i} \cdot E(\Delta W_i) = 0 \ (i \neq j)$$

*as is the case also for the cross products between $W_i$ and $n_j$ (all $i$, $j$), and $n_i$ and $n_j$ ($i \neq j$). Then, we can obtain*

$$E(\|\widetilde{\mathcal{F}}(X_0 + \Delta X) - \widetilde{\mathcal{F}}(X_0)\|^2) = \sum_{i=1}^{L} E\Big( \|\frac{\partial \widetilde{\mathcal{F}}}{\partial W_i} \cdot \Delta W_i\|^2 \Big) + \sum_{i=1}^{L} E\Big( \|\frac{\partial \widetilde{\mathcal{F}}}{\partial n_i} \cdot \Delta n_i\|^2 \Big). \quad (5)$$

*(5) is the result we want because, again, according to the Taylor series expansion (ignoring the higher order terms),*

$$\frac{\partial \widetilde{\mathcal{F}}}{\partial W_i} \cdot \Delta W_i = \widetilde{\mathcal{F}}(W_1, ..., W_i + \Delta W_i, ..., 0) - \widetilde{\mathcal{F}}(W_1, ..., W_L, 0, ..., 0),$$

$$\frac{\partial \widetilde{\mathcal{F}}}{\partial n_i} \cdot \Delta n_i = \widetilde{\mathcal{F}}(W_1, ..., W_L, ..., \Delta n_i, ..., 0) - \widetilde{\mathcal{F}}(W_1, ..., W_L, 0, ..., 0).$$

*Therefore, after dividing both sides of (5) by the dimensionality of the output vector of the neural network, the left side of (5) becomes $D_{\mathcal{F}}$ and the right side of (5) becomes the sum of all output error due to the quantization of individual layer's weights and activations.*

## 2 PARETO-OPTIMAL BIT ALLOCATION ON RESNET-50 AND MOBILENET-V1

Figure 2 shows the Pareto-optimal bit allocation of weights and activations on ResNet-50 He et al. (2016) and MobileNet-v1 Howard et al. (2017) when weights and activations are compressed to 2 bits and 4 bits on average. Results are generated on ImageNet dataset Deng et al. (2009). Two observations stand out on both ResNet-50 He et al. (2016) and MobileNet-v1 Howard et al. (2017). First, weights receive much larger bitrates (average bits per layer) than activations in general. Second, the layers with larger number of weights receive relatively lower bitrates; conversely, the layers with small number of weights receive relatively high bitrates.

The bit allocation pattern for weights is related to the variance of weights across layers. According to the classical Lagrangian rate-distortion formulation, the optimal bit allocation has positive correlation with the variance of the information source. Figure 3 shows layer-wise relationships between bitrate and variance of weights on ResNet-50 and MobileNet-v1. One can see that, for both ResNet-50 and MobileNet-v1, the layers with larger number of weights typically have smaller variances, and thus these layers receive smaller bitrates (and vice-versa).

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

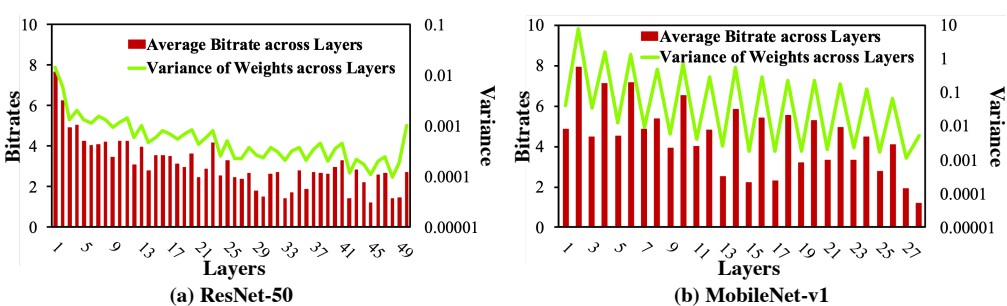

Figure 2: Pareto-optimal bit allocation of weights and activations across layers on ResNet-50 and MobileNet-v1 when weights and activations are compressed to 2 bit and 4 bits on average.

Figure 3: The relationship between the variances and the bitrates of weights across layers on ResNet-50 and MobileNet-v1 when compressed to 2 bits on average.

