# OpenReview forum: "Towards Effective 2-bit Quantization: Pareto-optimal Bit Allocation for Deep CNNs Compression"
_ICLR.cc/2020/Conference — Reject_

### Official Review · AnonReviewer3 · 2019-10-23
**Official Blind Review #3**

**Rating:** 8

**Review:**

Very good paper that studies the error rate of low-bit quantized networks and uses Pareto condition for optimization to find the best allocation of weights over all layers of a network. The theoretical claims are strongly supported by experiments, and the experimental analysis covers state-of-the-art architectures and demonstrates competitive results. The paper in addition also analyzes the inference cost of their approach (in addition to the accuracy results), and shows positive results on ResNet and MobileNet architectures.

The paper primarily shows that the mean squared error of the final output of a quantized network has the additive property of being equal to the sum of squared errors of the outputs obtained by quantizing each layer individually. Although there is no reason why this should be case, experimental results from the authors on AlexNet and VGG-16 validate this. Based on this assumption, the authors then use a Lagrangian based constrained optimization to minimize the sum of squared errors of outputs when individual weights/activations are quantized, with the constraint being the total bit budget for weights and activations. The authors show that this can be optimized under the Pareto condition easily.

The experimental section is quite detailed and covers the popular architectures instead of toy ones. The accuracy results compared to other 2-bit and 4-bit approaches are competitive. It's also nice to see analysis of inference cost where unequal bitrate allocation performs better than other methods.

The authors show that given the constrained optimization, layers that have a large number of weights receive lower bitrates and vice-versa. While it makes sense that this would contribute to stronger inference speedup compared to methods with either equal bitrate allocation across layers or those that allocate higher bitrate to layers with large number of weights, it's not entirely clear why the optimization would produce this allocation in the first place. Do the authors mean to conclude that layers with large number of weights hold a lot of redundancy and don't have a significant impact on the overall accuracy of the model? This needs to be clarified further.

**Experience Assessment:**

I have read many papers in this area.

**Review Assessment: Checking Correctness Of Derivations And Theory:**

I assessed the sensibility of the derivations and theory.

**Review Assessment: Checking Correctness Of Experiments:**

I assessed the sensibility of the experiments.

**Review Assessment: Thoroughness In Paper Reading:**

I read the paper at least twice and used my best judgement in assessing the paper.

---

> ### Author Response · Authors · 2019-11-09
> **Response to Reviewer #3 Questions**
>
>
> Thank you for the careful reviews and for the comments. We answer your questions below.
>
> Q1: Do the authors mean to conclude that layers with large number of weights hold a lot of redundancy and don't have a significant impact on the overall accuracy of the model? This needs to be clarified further.
>
> We empirically found that the layers having a larger number of weights receive lower bitrates (and vice-versa). The reason could be the values of the variances of the layers. According to the classical Lagrangian rate-distortion formulation, the optimal bit allocation follows a rule,
>
> Rate = G( - 1 / sigma^2 ),
>
> where G(.) is a strictly increasing function and sigma^2 is the variance of the variables. Based on the rule above, the layers with larger variances receive larger bitrates (and vice-versa).
>
> We calculated the variances of layers for two deep networks ResNet-50 and MobileNet-v1 (see Table 1 and Table 2 below). The results show that the layers with a smaller number of weights typically have a larger variance, and thus these layers receive larger bitrates. We added a paragraph and a figure in the supplementary material to discuss the relationship between variances and bitrates. The draft has been updated accordingly. Thank you for this good question.
>
> Table 1 – Variances of Weights across Layers on ResNet-50
> +-------------------------+----------+----------+----------+-----------+------------+
> |      Layer Index      |      5     |      15   |     25   |      35     |      45     |
> +-------------------------+----------+----------+----------+-----------+------------+
> | #Weights (10^5)   |  0.16   |    1.47  |   2.62  |    2.62    |    23.6    |
> +-------------------------+----------+----------+----------+-----------+------------+
> |    Variance (10^3) |  1.33    |    0.59  |   0.51   |    0.37   |    0.11    |
> +-------------------------+----------+----------+----------+-----------+------------+
> |        Bitrate             |    4.2    |    3.5   |     3.3   |     2.8    |     1.2      |
> +-------------------------+----------+----------+----------+-----------+------------+
>
> Table 2 – Variances of Weights across Layers on MobileNet-v1
> +-------------------------+----------+----------+----------+-----------+------------+
> |      Layer Index      |      2     |      8     |    14     |     20     |      26     |
> +-------------------------+--------- +----------+----------+-----------+------------+
> | #Weights (10^7)   |   0.29  |   1.15   |   4.61   |   4.61    |     9.22   |
> +-------------------------+----------+----------+----------+-----------+------------+
> |      Variance            |   7.83  |    0.49  |  0.54   |   0.23    |    0.07     |
> +-------------------------+----------+----------+----------+-----------+------------+
> |       Bitrate              |   7.9    |   5.38   |   5.86   |  5.29     |    4.1       |
> +-------------------------+----------+----------+----------+-----------+------------+

---

### Official Review · AnonReviewer2 · 2019-10-24
**Official Blind Review #2**

**Rating:** 6

**Review:**

This work nicely proposes a new theoretically-sound unequal bit allocation algorithm, which is based on the Lagrangian rate-distortion formulation. Surprisingly, the simple Lagrange multiplier on the constraints leads us to the convenient conclusion that the rate distortion curves for the weight quantization and the activation quantization have to match. Based on this conclusion, the authors claim that their search for the best bit allocation strategy is with a less complexity.

I found this paper interesting and enjoyed reading it. However, I wish the paper could address some issues that are a little bit confusing to me.

First of all, the paper is not about 2bit quantization. It seeks an “average” 2bit quantization. It means that some weights in some layers can be quantized with higher or lower bits per weight.  Same story goes on for the activation quantization. I don’t exactly know the implication of this, but it seems that the hardware implementation of a convolution layer could be either too complicated to benefit from this quantization scheme, or doesn’t really improve the efficiency of, say 4bit quantization for all layers. Is it really more efficient to do multiplication-and-addition between 2 bit weights and 5 bit input (the output of the previous activation) than between 4bit weights and 4bit input? I’m not a hardware person, but this part needs to be clearly addressed. Storage-wise, lowering the bitrate might be a clear benefit (I guess)

I wish the actual optimization part briefly mentioned in section 4.2 could be elaborated more. It is a crucial part but somewhat understated.

I also wonder what’s the effect or limitation of using MSE for this optimization, where cross-entropy is a more suitable choice. I know that the objective function in eq 5 is just to find the best combination of bit allocations per layer, but still, the error space might not be the best for this classification problem.

**Experience Assessment:**

I have published one or two papers in this area.

**Review Assessment: Checking Correctness Of Derivations And Theory:**

I assessed the sensibility of the derivations and theory.

**Review Assessment: Checking Correctness Of Experiments:**

I assessed the sensibility of the experiments.

**Review Assessment: Thoroughness In Paper Reading:**

I read the paper at least twice and used my best judgement in assessing the paper.

---

> ### Author Response · Authors · 2019-11-08
> **Response to Reviewer #2 Comments**
>
> Thank you for the careful reviews and for the comments. We answer your questions below.
>
>
> Q1: First of all, the paper is not about 2bit quantization. It seeks an “average” 2bit quantization. … Is it really more efficient to do multiplication-and-addition between 2 bit weights and 5 bit input (the output of the previous activation) than between 4bit weights and 4bit input?
>
> We did a comparison of the computational complexity of our method and the equally quantized method on ResNet-50. We calculated the number of arithmetic operations of both methods required to perform single inference.
>
> Specifically, we define a 32-bit multiplication/addition operation as one operation. To count the number of operations of the mixed-precision computation (e.g., 3 bit weight and 5 bit input), we follow the protocol defined by MicroNet challenge (https://micronet-challenge.github.io/scoring_and_submission.html) and consider the resolution of an operation to be the maximum bit-width of the 2 operands of this operation. For example, a multiplication operation with one 3-bit and one 5-bit operand will count as 5/32 of an operation.
>
> +-------------------------+-------------------+-------------------+--------------------+
> |            size              |        2 bits       |        4 bits       |         6 bits        |
> +-------------------------+-------------------+-------------------+--------------------+
> | equally quantized|   3.6 x 10^8    |   7.2 x 10^8    |    10.8 x 10^8  |
> +-------------------------+-------------------+-------------------+--------------------+
> |     our method      |   5.1 x 10^8    |   7.0 x 10^8    |    9.4 x 10^8     |
> +-------------------------+-------------------+-------------------+--------------------+
>
> The table above shows the amount of operations required when weights and activations are compressed to 2 bits, 4 bits and 6 bits respectively. Our unequally quantized method has less amount of operations than the equally quantized method when weights and activations are compressed to 4 bits and 6 bits on average. While, at 2 bits, our method has 1.4x more operations than the equally quantized method.
>
> On the other hand, the amount of operations does not imply equivalent inference speed in practice, as the processing of deep networks on hardware devices is constrained by both compute and memory access. We would like to reiterate that our method is effective to reduce the memory access time and thus provide higher inference rate compared to the equally quantized method, particularly for memory-bound hardware platforms where data movements are much slower and less energy efficient than compute. This is achieved by Pareto-optimal bit allocation which tends to allocate fewer bits per weight for layers that have a lot of weights. Thus, given fixed bandwidth, more weights can be loaded from off-chip memory to on-chip memory when processing the layers with a lot of weights, which in turn reduces compute idle time and improves the overall inference rate.
>
> To verify the point above, we simulated the inference speed on Google TPU v1 at 2 bits for both equally and unequally quantized methods with ResNet50. As existing hardware does not well support mixed-precision operations, we assume the weights and activations with unequal bit-widths are fetched from off-chip memory, then decoded to fixed 8-bit stream and fed to compute unit that supports 8-bit multiplications (e.g. TPU). The simulation results on Google TPU platform show that our method is 1.5x faster than the equally quantize method.
>
> We will add a paragraph to clarify the computational complexity of our method and the equally quantized method.

---

> > ### Author Response · Authors · 2019-11-08
> > **Response to Reviewer #2 Comments (part 2)**
> >
> >
> > Q2: I wish the actual optimization part briefly mentioned in section 4.2 could be elaborated more. It is a crucial part but somewhat understated.
> >
> > Thank you for this suggestion. We will elaborate section 4.2 to show the optimization steps in more details. We will respond to this comment again once we finish the revision.
> >
> >
> > Q3: I also wonder what’s the effect or limitation of using MSE for this optimization, where cross-entropy is a more suitable choice. I know that the objective function in eq 5 is just to find the best combination of bit allocations per layer, but still, the error space might not be the best for this classification problem.
> >
> > We agree that MSE does not directly optimize accuracy and thus may not be the best choice for classification problem. The reason we choose MSE as the measurement of the quantization impact is mainly because it ensures that the additivity property of output error holds, from both empirical observations and mathematical derivations (as shown in the draft), and the additivity property is essential for Pareto condition. Besides, optimization with MSE not only supports classification tasks but also can be applied to any other tasks like object detection and semantic segmentation where regression loss is also required.
> >
> > +-----------------------+-----------+----------+-----------+-----------+
> >  |            size          |  4 bits   | 6 bits  |  8 bits   | 10 bits  |
> > +-----------------------+-----------+----------+-----------+-----------+
> >  |  cross-entropy  |    41.3   |  43.6    |   46.0    |   57.0    |
> > +-----------------------+-----------+----------+-----------+-----------+
> >  |   MSE                  |   63.6    |   70.8   |    70.9   |   70.9    |
> > +-----------------------+-----------+----------+-----------+-----------+
> >
> > Cross-entropy is a more suitable choice for classification, but our empirical observations show that it is not compatible with the Pareto optimal bit allocation framework. The table above shows the results on MobileNet-v1 when replacing MSE in Eq. 3 with cross-entropy for optimization. One can see that there is a noticeable accuracy drop using cross-entropy in the optimization framework. We also observed that the additivity property doesn’t hold anymore if we use cross-entropy as the measurement. From the mathematical point of view, it is unclear whether or not the additivity property is still valid for metrics beyond MSE, we would like to leave it for future study. Thank you for this insight.

---

> ### Author Response · Authors · 2019-11-13
> **Revisions to the Draft**
>
> We revised the draft based on the comments:
> -	We added section 6.1 to clarify the computational complexity of our method.
> -	We made a comparison of the amount of arithmetic operations to the equally quantized method in section 6.1.
> -	The optimization method is elaborated in section 4.2. A figure is also added to show an example of the optimization method (Fig. 3).
> The updated version has been uploaded.

---

> ### Comment · AnonReviewer2 · 2019-11-15
> **Acknowledge the rebuttal**
>
> I appreciate the authors' responses to my questions. I can see that the paper now considers the efficiency issue of the mixed-precision feedforward more fairly.

---

### Official Review · AnonReviewer1 · 2019-10-27
**Official Blind Review #1**

**Rating:** 1

**Review:**

This works presents a method for inferring the optimal bit allocation for quantization of weights and activations in CNNs. The formulation is sound and the experiments are complete. My main concern is regarding the related work and experimental validation being incomplete, as they don't mention a very recent and similar work published in ICIP19 https://ieeexplore.ieee.org/document/8803498: "Optimizing the bit allocation for compression of weights and activations of deep neural networks". A reference in related work as well as a comparison in experimental validation would be necessary  and the novelty of this work is rather weak given the above mentioned 2019 publication.

**Experience Assessment:**

I have published one or two papers in this area.

**Review Assessment: Checking Correctness Of Derivations And Theory:**

I assessed the sensibility of the derivations and theory.

**Review Assessment: Checking Correctness Of Experiments:**

I assessed the sensibility of the experiments.

**Review Assessment: Thoroughness In Paper Reading:**

I read the paper at least twice and used my best judgement in assessing the paper.

---

> ### Author Response · Authors · 2019-11-07
> **Clarification - the Main Differences with the ICIP paper**
>
>
> Thank you for your reviews and for pointing out the ICIP paper. We notice that the ICIP paper was posted on IEEE Xplore website on 22 Sept, which is 3 days prior to the ICLR deadline (Sept 25). According to the ICLR 2019 reviewer guideline, “no paper will be considered prior work if it appeared on arxiv, or another online venue, less than 30 days prior to the ICLR deadline”. We believe that our submission meets the ICLR regulations and rules.
>
> Our ICLR submission has substantial differences with the mentioned ICIP paper including the theoretical analysis, methods and insights, and experimental results. Moreover, with the new compression framework, the ICLR submission achieves 2-bit quantization results on deep architecture ResNet-50. To our best knowledge, this is the first work that reports 2-bit results without hurting the accuracy. Below we summarize the main differences with the ICIP paper:
>
> (1) Our ICLR submission provides a mathematical derivation for the additivity property. With two reasonable assumptions, we demonstrate that the additivity property holds for any neural networks which are continuously differentiable in the layers.
>
> (2) Our quantization framework differs from the ICIP paper in two-fold. First, we adopt a dead zone to the quantization function of weights. Second, we apply the straight-through estimator (STE) to perform back-propagation on the retraining stage for both quantized weights and activations. The ICIP paper uses the simple uniform quantizer and the framework does not provide a scheme to support the retraining for quantized weights and activations. However, as we illustrated in the experiment section, dead zone and STE retraining are critical for improving the accuracy.
>
> (3) In our ICLR submission, we reveal that the pattern of Pareto-optimal bit allocation across layers has positive impacts on neural network inference rate in practice. It tends to allocate fewer bits per weight for layers that have a lot of weights, which helps to reduce memory-access time which in turn reduces compute idle time and improves the overall inference rate. We verified this point by designing hardware simulation experiments on Google TPU v1 platform. Results show that the Pareto-optimal bit allocation improves the inference rate on ResNet50 by 1.5x compared to its equal bit allocation counterpart.
>
> (4) Combined the dead-zone quantization and STE based retraining with the optimal bit allocation strategy, our quantization framework achieves state-of-the-art result on deep neural network ResNet-50 at 2 bits. To the best of our knowledge, this is the first work that reports 2-bit results without hurting the accuracy. The ICIP paper can only compress ResNet-50 down to 4 bits and the accuracy drops significantly at 2 bits.
>
> We will change our ICLR draft accordingly, and then upload it to the review website. We would also like to answer any other questions.

---

> ### Author Response · Authors · 2019-11-13
> **Revisions to the Draft**
>
> We revised the draft based on the comments:
> -	We cited the ICIP paper and discussed the differences with the ICIP paper in the related work section.
> -	We added the results of the ICIP paper in Table 1 and compared our method with the ICIP paper in the experiment section.
> The updated version has been uploaded.

---

### Public Comment · ~Evgenii_Zheltonozhskii1 · 2019-09-27
**Strong results, but some concerns about low precision arithmetics**

Nice work. The result you have achieved are really spectacular.
In the simulations you assume that all calculations are performed in full precision available to the accelerator (8 or 16 bits), while for custom hardware it would make more sense to use lower-precision arithmetic, which can provide significant speedups. However, I have two concerns regarding application of this approach to the proposed method which I hope you can address:

1. Uniformity of quantization.
The main advantage of uniform quantization is the fact it is isomorphic, which allows to apply integer arithmetic to operate over bin indices rather than values themselves. If I understand correctly, dead zone quantization, unfortunately, lacks this property, meaning it would require some more complicated implementation to perform matrix multiplication, for example, using lookup tables.
Possibly it can be done by taking account of beta separately, but it's not 100% clear for me and might introduce additional computational overhead.
2. Computational complexity vs. memory requirements.
Most of work you compare to, do not perform additional compression except quantization. That means, the number of bits provided by those works applies both to computational complexity and memory requirements. On the other hand, since your work performs additional compression of the weights and activation, in your case those numbers are not equal anymore. From my understanding, provided numbers are the average amount of storage required for one value (of either weights or activations). That would probably mean that amount of computation required for the network would not be equivalent to an equally quantized two-bit network.
Could you provide some number regarding computational requirements of inference for your method (for example amount of bit-operations required for single inference)?

---

> ### Author Response · Authors · 2019-11-08
> **Response to Question 1 and 2**
>
> Thank you for your interest in this paper and for your comments. Below we answer your questions.
>
>
> Q1: Uniformity of quantization
>
> Dead zone quantization can also support integer arithmetic if we set the values of the first negative and the first positive quantization centroids as k * delta, where k is an integer value and delta is the length of quantization interval. By doing this, every quantization centroid becomes an integral multiple of delta and we can use the integral multiplier for integer arithmetic.
>
> For example, if we set k = 2, the corresponding quantization centroids are:
>
>  ... , -n * delta , ... , -3 * delta , -2 * delta , 0 , 2 * delta , 3 * delta , ... , n * delta , ...
>
> and the integral multipliers "... , -n , ... , -3 , -2 , 0 , 2 , 3 , ... , n , ..." can be used for integer arithmetic.
>
> We will introduce how to apply integer arithmetic with dead zone quantization and update the paper accordingly. Thank you for this good point.
>
>
> Q2: Computational complexity vs. memory requirements
>
> We calculated the amount of arithmetic operations of our method required to perform a single inference on ResNet-50, and did a comparison with the equally quantized method (please see our response to Q1 of Reviewer #2). The results show that our method has fewer operations than the equally quantized method at 4 bits and 6 bits. While, at 2 bits, our method has 1.4x more operations than the equally quantized method.
>
> In practice, the inference time on hardware devices is constrained by both compute and memory access. The Pareto-optimal bit allocation tends to allocate fewer bits per weight for layers that have a lot of weights. As a result, it helps to reduce the corresponding memory-access time which in turn reduces compute idle time and improves the overall inference speed. The simulation results on the Google TPU platform show that our method is 1.5x faster than the equally quantize method on ResNet-50 at 2 bits. Please see our response to Reviewer #2 for the details.

---

### Public Comment · ~Kevin_Zhang2 · 2019-09-28
**No code in provided dropbox link even after 56 hours of submission deadline**

Hi,
As of close to 56 hours after submission deadline , no code is present in the provided dropbox link. It is not fair to provide a placeholder link for code submissions (which impact the review process) and submit code taking considerable buffer time after submission deadline.

---

> ### Author Response · Authors · 2019-09-30
> **Codes are available**
>
> Thanks for your interest in our work. Apologies that we didn't have enough time to clean up our code before the deadline, also, we felt it is necessary to double-check the quantized models and redo evaluations again to make sure all the results reported here are reproducible.
>
> Finally, we have uploaded the evaluation code and quantized models to the dropbox link provided earlier. Kindly let us know if there is any question regarding the code.

---

> > ### Public Comment · ~Saeed_Ranjbar1 · 2019-10-05
> > **code is not available for OPTIMIZATION UNDER PARETO CONDITION**
> >
> > Dear Authors,
> > Thanks for your comment. From review of your code, I cannot see the code related to your optimization section. The details regarding how you calculated the slopes and how you considered the rate constraints are missing.

---

### Author Response · Authors · 2019-11-13
**Summary of Revisions**


We thank all reviewers for their careful reviews, insightful comments and feedback on our paper. The draft has been revised accordingly. The revised draft has been uploaded. The main revisions are summarized below:

-	We cited the ICIP paper mentioned by Reviewer 1 and discussed the differences with the ICIP paper in the related work section. We added the results of the ICIP paper in Table 1 and compared our method with the ICIP paper in the experiment section. (response to Reviewer 1)
-	We added a section (section 6.1) to clarify the computational complexity of our method. We made a comparison of the amount of arithmetic operations to the equally quantized method in section 6.1. (response to Reviewer 2)
-	The optimization method is elaborated in section 4.2. A figure is also added to show an example of the optimization method (Fig. 3). (response to Reviewer 2)
-	We discussed the relationship between the variances and the bitrates in the supplementary material. (response to Reviewer 3)

Here we summarize the main differences with the ICIP paper mentioned by Reviewer 1:

-	Our ICLR submission shows the additivity property of the output error and provides a mathematical derivation for the additivity property.
-	Our quantization framework differs from the ICIP paper in two-fold. First, we adopt a dead zone to the quantization function of weights. Second, we provide a scheme to support the retraining for both quantized weights and activations.
-	Our work demonstrates that the optimal bit allocation solved by our method has very positive impacts on inference speed, which has been verified by the hardware simulation experiments.
-	By combining the dead-zone quantization and STE based retraining with the optimal bit allocation strategy, our quantization framework achieves state-of-the-art results. To the best of our knowledge, our work is the first to report 2-bit results on deep neural network ResNet-50 without hurting the accuracy on ImageNet.

Best,
Authors

---

### Decision · Program_Chairs · 2019-12-19

**Decision:**

Reject

**Comment:**

This works presents a method for inferring the optimal bit allocation for quantization of weights and activations in CNNs. The formulation is sound and the experiments are complete. However, the main concern is that the paper is very similar to a recent work by the authors, which is not cited.